# *Rubinosporus**auriporus* gen. et sp. nov. (Boletaceae: Xerocomoideae) from Tropical Forests of Thailand, Producing Unusual Dark Ruby Spore Deposits

**DOI:** 10.3390/jof8030278

**Published:** 2022-03-09

**Authors:** Santhiti Vadthanarat, Olivier Raspé, Saisamorn Lumyong

**Affiliations:** 1Department of Biology, Faculty of Science, Chiang Mai University, Chiang Mai 50200, Thailand; santhiti.research@mfu.ac.th; 2Research Center of Microbial Diversity and Sustainable Utilization, Faculty of Science, Chiang Mai University, Chiang Mai 50200, Thailand; 3School of Science, Mae Fah Luang University, Chiang Rai 57100, Thailand; 4Academy of Science, The Royal Society of Thailand, Bangkok 10300, Thailand

**Keywords:** fungal diversity, multigene phylogeny, new genus, taxonomy

## Abstract

*Rubinosporus*, a new bolete genus from tropical forests of Thailand is introduced with *R*. *auriporus* as the type species. The genus is unique among Xerocomoideae in producing dark ruby spore deposits. It can be differentiated from all other Boletaceae genera by the following combination of characters: pileus surface evenly covered with matted tomentum; stipe surface with evenly scattered minute squamules; golden yellow tubular hymenophore, which is relatively thin especially when young; unchanging surfaces and context when bruised or cut; smooth, broadly ellipsoid basidiospores; and dark ruby spore deposits. The Boletaceae-wide and Xerocomoideae-wide phylogenetic analyses based on four-gene data sets (*atp*6, *cox*3, *rpb*2, and *tef*1) support *Rubinosporus* as monophyletic and places it in Boletaceae subfamily Xerocomoideae. Full descriptions and illustrations of the new genus and species are presented.

## 1. Introduction

The family Boletaceae has been widely studied for over one hundred years. The former traditional taxonomy was based only on morphological characters. However, since molecular techniques and phylogenetic analyses have been developed and used as an advanced tool for the modern concepts in systematics and taxonomy, many genera, and species in Boletaceae have been recognized and described as new, e.g., [1,2,3]. In the last five years only, nine new Boletaceae genera have been described worldwide, namely *Afrocastellanoa* M.E. Smith & Orihara, *Cacaoporus* Raspé & Vadthanarat, *Carolinigaster* M.E. Sm. & S. Cruz, *Erythrophylloporus* Ming Zhang & T.H. Li, *Indoporus* A. Parihar, K. Das, Hembrom & Vizzini, *Ionosporus* O. Khmelnitsky, *Phylloporopsis* Angelini, A. Farid, Gelardi, M.E. Smith, Costanzo, & Vizzini, *Spongispora* G. Wu, S.M.L. Lee, E. Horak & Zhu L. Yang, and *Longistriata* Sulzbacher, Orihara, Grebenc, M.P. Martín & Baseia [4,5,6,7,8,9,10,11,12]. Five of those genera were described from tropical to subtropical Asia, where high fungal diversity has been reported, but yet, remains poorly known to science, e.g., [13,14].

Based on the current multiple gene phylogenies, the Boletaceae are classified into six sub-families and one phylogenetically unsupported group [2,3]. Xerocomoideae is one of the six sub-families, which consists of nine genera namely *Alessioporus* Gelardi, Vizzini & Simonini, *Aureoboletus* Pouzar, *Boletellus* Murrill, *Heimioporus* E. Horak, *Hemileccinum* Šutara, *Hourangia* Xue T. Zhu & Zhu L. Yang, *Phylloporus* Quél., *Pulchroboletus* Gelardi, Vizzini & Simonini, and *Xerocomus* Quél [2,3]. Two additional genera, *Corneroboletus* and *Sinoboletus*, were also described in the subfamily but were later synonymized with *Hemileccinum* and *Aureoboletus*, respectively [3]. Typical characters of this subfamily are boletoid or phylloporoid basidiomata; dry or viscid pileus with smooth or subtomentose to tomentose pellis; absent or rarely present veil; yellowish to yellow context; often bluing or sometimes redding or unchanging; smooth or ornamented stipe surface; basidiospores with bacillate, reticulate, longitudinally striate, or pitted ornamentations, or occasionally smooth; spore deposit with more or less olive-brown tint [2,3,15].

We have carried out surveys of the diversity of boletes in Thailand since 2010. Some collections with striking morphological characters were made and carefully studied. The collections combined typical characters of two genera that are widely distributed in tropical to subtropical regions, *Aureoboletus* with a golden yellow hymenium, and *Baorangia* G. Wu & Zhu L. Yang, which has a thin hymenophoral layer [3,16,17]. However, the collections showed a surprising dark ruby spore deposit, which is clearly distinct from the two genera and other genera in Boletaceae. Therefore, family-wide and subfamily-wide phylogenies were performed and showed that the collections belong in a generic lineage different from other genera in Boletaceae. Consequently, a new genus and a new species are introduced, with full descriptions and illustrations.

## 2. Materials and Methods

### 2.1. Specimens Collecting

The specimens were collected during the rainy season, from May to June, between 2015 and 2017, in Chiang Mai Province, northern Thailand. The specimens were wrapped in aluminum foil and taken to the laboratory for morphological description. After the description of macroscopic characters, the specimens were dried in an electric drier at 45–50 °C for 24 h or until dried properly. Then, they were deposited in the following herbaria Chiang Mai University (CMUB), and Meise Botanic Garden (BR) [18].

### 2.2. Morphological Study

The macroscopic descriptions were made based on detailed field notes and photos of fresh basidiomata taken in the habitat and the laboratory. Color codes were given based on a Methuen Handbook of Colour [19]. Chemical solutions including 10% potassium hydroxide (KOH) and 28–30% ammonium hydroxide (NH_4_OH), were used to determine the chemical reactions (color reactions) of the pileus, pileus context, stipe, stipe context, and hymenophore. For the microscopical study, the dried specimens were observed using 5% KOH, NH_4_OH, Melzer’s reagent, or stained with 1% ammoniacal Congo red. A minimum of 50 basidiospores, 20 basidia, and 20 cystidia were randomly measured under a Nikon Eclipse Ni microscope using the NIS-Elements D software. The notation ‘[m/n/p]’ represents the number of basidiospores “m” measured from “n” basidiomata of “p” collections. Dimensions of microscopic structures are presented in the following format: (a–)b–c–d(–e), in which “c” represents the average, “b” the 5th percentile, “d” the 95th percentile, “a” the minimum, and “e” the maximum. Q, the length/width ratio, is presented in the same format. Section of the pileus surface was radially and perpendicularly cut to the surface at a point halfway between the center and margin of the pileus. Sections of stipitipellis were taken from halfway up the stipe and longitudinally cut perpendicularly to the surface. All microscopic features were drawn by freehand using an Olympus Camera Lucida model U−DA fitted to Olympus CX31 compound microscope. For scanning electron microscopy (SEM), a spore print was mounted onto an SEM stub with double-sided carbon tape. The sample was coated with gold, then examined and photographed with a JEOL JSM–5910 LV SEM (JEOL, Tokyo, Japan).

### 2.3. DNA Extraction, PCR Amplification and DNA Sequencing

Genomic DNA was extracted from about 10–15 mg of dried specimen or fresh tissue preserved in cetyltrimethylammonium bromide (CTAB), using a CTAB isolation procedure adapted from Doyle and Doyle [20]. Portions of the genes *atp*6, *cox*3, *rpb*2, and *tef*1 were amplified by polymerase chain reaction (PCR) and sequenced by Sanger sequencing. The primer pairs ATP6-1M40F/ATP6-2M [21], COX3M1-F/ COX3M1-R [11], bRPB2-6F/bRPB2-7.1R [22], and EF1-983F/EF1-2218R [23] were used to amplify *atp*6, *cox*3, *rpb*2, and *tef*1, respectively. PCR products were purified by adding 1 U of exonuclease I and 0.5 U FastAP alkaline phosphatase (Thermo Scientific, St. Leon-Rot, Germany) and incubated at 37 °C for 1 h, followed by inactivation at 80 °C for 15 min. Standard Sanger sequencing was performed in both directions by Macrogen with PCR primers, except for *atp*6, for which universal primers M13F-pUC(−40) and M13F(−20) were used; for *tef*1, additional sequencing was performed with two internal primers, EF1-1577F and EF1-1567R [23].

### 2.4. Alignment and Phylogeny Inference

The two reads of newly generated sequences were assembled in GENEIOUS Pro v. 6.0.6 (Biomatters). A Boletaceae-wide sequence dataset, including selected sequences representative of the whole family, downloaded from GenBank, was aligned using MAFFT [24] on the server accessed at http://mafft.cbrc.jp/alignment/server/ (accessed on 19 December 2021). For this dataset, the introns in *rpb*2 and *tef*1 were removed based on the amino acid sequence of previously published sequences. Maximum likelihood (ML) phylogenetic inference was performed using RAxML on the CIPRES web portal (RAxML-HPC2 on XSEDE) [25,26]. The phylogenetic tree was inferred by a single analysis with four partitions (one for each gene), using the general time reversible computerized adaptive testing (GTRCAT) model with 25 categories. The outgroup consisted of two *Buchwaldoboletus* and seven *Chalciporus* species from sub-family Chalciporoideae, based on previous phylogenies e.g., [1,2,3,11]. Statistical support of clades was obtained with 1000 rapid bootstrap replicates. For Bayesian Inference (BI), the best-fit model of substitution among those implementable in MrBayes was estimated separately for each region using jModeltest [27] on the CIPRES portal, based on the Bayesian Information Criterion (BIC). The selected models were HKY + I + G for atp6, GTR+I+G for cox3 and tef1 exons, and K80 + I + G for rpb2 exons. Partitioned Bayesian analysis was performed with MrBayes 3.2.6 software for Windows [28]. Two runs of five chains were run for 11,000,000 generations and sampled every 1000 generations. The chain temperature was decreased to 0.02 to improve convergence. At the end of the run, the average deviation of split frequencies was 0.008614. A total of 8252 trees were used to construct a 50% majority rule consensus tree and calculate the Bayesian posterior probabilities (BPP).

For a subfamily Xerocomoideae-wide tree, all selected taxa in Xerocomoideae were aligned using the MAFFT online software (introns included). ML phylogenetic tree was inferred by a single analysis with five partitions (*atp*6, *cox*3, *rpb*2 exons, *tef1* exons, and intron of *rpb*2 + introns of *tef*1) (one for each gene), outgroup were four *Butyriboletus* species in *Pulveroboletus* group, using the same analytical software and model used for family Boletaceae-wide tree. For BI, the same analytical software for family Boletaceae-wide tree was used. However, the selected models were GTR+I+G for *atp*6, *cox*3, and intron of *rpb*2 + introns of *tef*1, K80 + I + G for *rpb*2 exons, and SYM+I+G for *tef*1 exons. Two runs of five chains were sampled every 200 generations and stopped after 800,000 generations. At the end of the run, the average deviation of split frequencies was 0.007928. A total of 2709 trees were used to construct a 50% majority rule consensus tree and calculate the BPPs.

## 3. Results

### 3.1. Phylogenetic Analyses

A total of fourteen sequences were newly generated in this study and deposited in GenBank (Table 1). For the Boletaceae-wide tree, the alignment contained 776 sequences comprising four genes (162 for *atp*6, 133 for *cox*3, 231 for *rpb*2, 250 for *tef*1) from 257 voucher specimens corresponding to 252 taxa, and was 2946 characters long (TreeBase number: 28,349). The sequences of *Rubinosporus* voucher SV0934 (*atp*6 and *cox*3) were not added to the analyses because they were identical to the holotype (SV0090). Maximum likelihood and BI trees of the combined four-gene dataset were similar in topology, without any supported conflict (BS ≥ 70% and PP ≥ 0.90). The phylogram of RAxML bipartition (Figure 1) retrieved the six subfamily clades, namely Austroboletoideae (BS = 99% and PP = 1), Boletoideae (BS = 54% and PP = 0.93), Chalciporoideae (BS = 100% and PP = 1), Leccinoideae (BS = 99% and PP = 1), Xerocomoideae (BS = 99% and PP = 1), and Zangioideae (BS = 100% and PP = 1). The *Pulveroboletus* group of Wu et al. [2,3] was not monophyletic, like in previously published phylogenies. However, the monophyly of each genus in this group was highly supported. The selected *Rubinosporus auriporus* specimens were monophyletic (BS = 100% and PP = 1) and clustered in the highly supported Xerocomoideae clade.

The Xerocomoideae-wide alignment contained 243 sequences comprising four genes (42 for *atp*6, 38 for *cox*3, 81 for *rpb*2, 82 for *tef*1) from 86 voucher specimens corresponding to 82 taxa and was 3176 characters long (TreeBase number: 28350). ML and BI trees showed similar topologies without any supported conflict. The phylogram of RAxML bipartition (Figure 2) retrieved nine highly supported generic clades, for which BS = 100% and PP = 1 for six clades, *Aureoboletus*, *Pulchroboletus*, *Heimioporus*, *Hemileccinum*, *Hourangia*, and the new genus *Rubinosporus*, while the others had only slightly less support, *Boletellus* (BS = 85% and PP = 1), *Phylloporus* (BS = 99% and PP = 1), and *Xerocomus* (BS = 75% and PP = 1).

### 3.2. Taxonomy

***Rubinosporus*** Vadthanarat, Raspé & Lumyong, **gen. nov.**

Typus generis—*Rubinosporus auriporus* Vadthanarat, Raspé & Lumyong

MycoBank—MB840262

Etymology—from Latin “rubineus” and “sporus” referring to its production of dark ruby spore deposits.

Diagnosis—Distinguished from the other genera in Boletaceae by the following combination of characters: pileus surface even, with matted, cracked tomentum; stipe surface even, scattered with minute squamules, golden yellow tubular hymenophore which is relatively thin, especially when young; unchanging surfaces and context when touched or cut; smooth, broadly ellipsoid basidiospores; dark ruby spore deposit.

Description—Basidiomata stipitate-pileate with tubular hymenophore, medium-sized. Pileus hemispherical at first then convex to plano-convex or applanate in age; margin inflexed to deflexed, exact to slightly exceeding; surface even to subrugulose at places, dull, greyish red to pastel red to reddish brown, with greyish yellow, greyish orange to brownish orange to brown matted, cracked tomentum; context firm, off-white to yellowish white, unchanging when cut. Stipe central, terete, or sometimes slightly compressed, cylindrical or subcylindrical with slightly wider base; surface topography even, yellowish white to pinkish white at places, with scattered yellowish white to orange to light brown minute squamules, to bright yellow near the top; basal mycelium yellowish white; context solid, off-white to yellowish white, unchanging when cut. Hymenophore tubulate, narrowly adnate, mostly segmentiform to subventricose. Tubes relatively thin, especially when young, golden yellow becoming orange-yellow, separable from the pileus context, unchanging when bruised. Pores topography subirregular, irregularly arranged, roundish to slightly angular composite pores; golden yellow at first, golden yellow to greyish yellow with irregularly reddish brown at places in age, unchanging when touched. Odor mild fungoid. Taste mild to slightly sweet. Spore print dark ruby in mass. Basidiospores broadly ellipsoid, thin-walled, smooth, yellowish to brownish hyaline in water, yellowish hyaline in KOH or NH_4_OH, yellowish to reddish in Melzer’s reagent (weakly dextrinoid). Basidia 4-spored, clavate without basal clamp connection. Cheilocystidia clavate with rounded apex or fusiform to broadly fusiform or utriform, thin-walled, hyaline to yellowish hyaline in KOH or NH_4_OH. Pleurocystidia fusiform with narrower apex, thin-walled, hyaline to yellowish hyaline in KOH or NH_4_OH. Pileipellis a tomentum to intricate trichoderm, composed of moderately interwoven thin-walled hyphae; terminal cells cylindrical with obtuse apex, hyaline to yellowish at places in KOH. Stipitipellis a tomentum composed of loosely to moderately interwoven cylindrical hyphae, anastomosing at places, scattered with groups of rising cells to clusters of basidiole-like cells mixed with caulocystidia, and rarely with caulobasidia, hyaline to yellowish hyaline in KOH or NH_4_OH. Clamp connections were not seen in any tissue.

Known distribution—Currently known only from Thailand.

Notes—The morphologically closely resembling genera are *Aureoboletus* and *Baorangia*, the former sharing the bright yellow to golden yellow hymenium, and the latter sharing the thin hymenophore [3,17,31]. However, *Rubinosporus* is easily distinguished from those two genera by the dark ruby spore deposit, which has an olive brown tint in *Aureoboletus* and *Baorangia*.

***Rubinosporus auriporus*** Vadthanarat, Raspé & Lumyong, **sp. nov.** Figure 3, Figure 4 and Figure 5

MycoBank—MB840263

Holotype—THAILAND, Chiang Mai Province, Mae Taeng District, 19°06’37.5” N–98°44’40.0” E, elev. 1,090 m, 2 June 2015, Santhiti Vadthanarat, SV0090 (CMUB; isotype BR).

Etymology—from Latin referring to the golden yellow hymenophore.

Description—Basidiomata medium-sized. Pileus (14–)33–92(–174) mm in diameter, hemispherical at first then convex to plano-convex or applanate in age; margin inflexed to deflexed, exact to slightly exceeding somewhere (0.5 mm); surface even to subrugulose at places, dull, greyish red to pastel red to reddish brown (8E5–6, 9B/C/D5–7, 9E6), at first densely to moderately covered with greyish yellow, greyish orange to brownish orange to brown (4B3–5, 5B3–5, 5C3–6, 5D3–4, 6E7–8) matted, cracked tomentum becoming less in age; context 4–15 mm thick half-way to the margin, firm, off-white to yellowish white (1A1–2), occasionally pale yellow (1A2–3) above the hymenium or under pileipellis, unchanging when cut. Stipe 37–109 × 11–45 mm, central, terete or slightly compressed sometimes, mostly cylindrical to cylindrical with slightly wider base; surface topography even, slightly shiny, yellowish white (2A2–3, 3A2) with pinkish white (8A2) at places, scattered with yellowish white (2A2–3, 3A2) to orange to light brown (5A/B7, 7D7–8) minute squamules, to bright yellow (2–3A7) near the top; basal mycelium little developed, yellowish white (2A2); context solid, off-white to yellowish white (1A1–2), even to virgate at places, unchanging when cut. Hymenophore tubulate, narrowly adnate, mostly segmentiform to subventricose. Tubes (0.8)2–4.5(7) mm long half-way to the margin, relatively thin when young 1/4 to 1/5 times then 1/2 to 1/3 times that of the pileus context when mature, golden yellow (3A7) becoming orange yellow (4B7), separable from the pileus context, unchanging when bruised. Pores 0.4–0.8(1) mm wide at mid-radius, topography subirregular, irregularly arranged, composite pores composed of roundish to slightly angular pores in age, golden yellow (3–4A8) at first, golden yellow to greyish yellow (4A/B/C7) with irregularly reddish brown (8E/F8) at places in age, unchanging when touched. Odor mild fungoid. Taste mild to slightly sweet. Spore print dark ruby (12F7) in mass.

Macrochemical reactions: KOH, yellow to orange on cap, stipe, and hymenium; none or yellowish on pileus context and stipe context; NH_4_OH, yellow to orange to brown on cap, stipe and hymenophore; none or yellowish on pileus context, stipe context and hymenium.

Spores [293/5/2] (6.5–)7.1–7.9–8.7(–9.3) × (4.4–)5.2–5.8–6.4(–6.9) µm Q = (1.19–)1.25–1.36–1.52(–1.68). From the type (7–)7.1–7.7–8.6(–9) × (4.4–)4.8–5.7–6.5(–6.8) µm, Q = (1.19–)1.23–1.36–1.53(–1.66), *N* = 60, broadly ellipsoid, thin-walled, smooth, yellowish to brownish hyaline in water, yellowish hyaline in KOH or NH_4_OH, yellowish to reddish in Melzer’s reagent (inamyloid to weakly dextrinoid). Basidia 4-spored, (18–)19–24–27(–28) × (9–)9–11–12(–12) µm, clavate without basal clamp connection, hyaline to yellowish hyaline in KOH or NH_4_OH; sterigmata up to 4 µm long. Cheilocystidia of two types, (1) clavate with rounded apex, frequent, (14–)15–25–36(–38) × (9–)10–12–16(–16) µm, thin-walled, hyaline to yellowish hyaline in KOH or NH_4_OH, and (2) fusiform to broadly fusiform or utriform, frequent, (21–)22–34–41(–41) × (10–)10–12–15(–16) µm, thin-walled, hyaline to yellowish hyaline in KOH or NH_4_OH. Pleurocystidia (29–)30–47–58(–61) × (9–)9–12–16(–18) µm, frequent and more near the pores, fusiform with narrower apex, thin-walled, hyaline to yellowish hyaline in KOH or NH_4_OH. Hymenophoral trama divergent, 57–106 µm wide, with 16–32 µm wide of subregular mediostratum, composed of cylindrical, 4–12 µm wide hyphae, slightly yellowish to hyaline in KOH or NH_4_OH. Pileipellis a tomentum to intricate trichoderm, 125–230 µm thick, composed of moderately interwoven thin-walled hyphae; terminal cells 21–68 × 3.5–9 µm, cylindrical with obtuse apex, hyaline to yellowish at places in KOH. Pileus context made of strongly interwoven, thin-walled, hyaline hyphae, 7–23 µm wide, hyaline in KOH. Stipitipellis a tomentum composed of loosely to moderately interwoven cylindrical hyphae (3–9 µm wide), anastomosing at places, scattered with groups of rising cells to clusters of basidiole-like cells ((14–)15–23–38(–39) × (5–)6–8–10(–11) µm) mixed with two types of caulocystidia, and rarely with caulobasidia, 120–170 µm thick (including the height of rising cells), hyaline to yellowish hyaline in KOH or NH_4_OH; terminal cells 24–81 × 5–9 µm, more or less parallel to the surface of the stipe, thin-walled, elongated cylindrical with obtuse to slightly swollen apex. Caulocystidia of two types, 1) fusiform, not frequent, (25–)26–45–72(–76) × (9–)9–14–18(–18) µm, thin-walled, hyaline to yellowish hyaline in KOH, and 2) broadly clavate, not frequent, (14–)14–23–34(–34) × (10–)10–15–21(–21) µm, thin-walled, hyaline to yellowish hyaline in KOH. Stipe context composed of parallel, 6–18(23) µm wide hyphae, hyaline to yellowish hyaline in KOH or NH4OH. Clamp connections were not seen in any tissue.

Habitat and distribution—Gregarious (up to 6 basidiomata) to fasciculate of 2–4 basidiomata, on soil in hill evergreen forest dominated by Fagaceae mixed with Dipterocarpaceae: *Dipterocarpus obtusifolius*, *D. costatus*, *Shorea siamensis*, *Hopea* sp. Currently known only from the type locality in Chiang Mai Province, northern Thailand.

Specimens examined—THAILAND, Chiang Mai Province, Mae Taeng District, 19°06’32.7” N–98°44’33.6” E, elev. 1,070 m, 4 Jun 2015, Santhiti Vadthanarat, SV0101 (CMUB, BR); ibid. 19°06’33.8” N–98°44’20.9” E, elev. 1110 m, 23 May 2017, Santhiti Vadthanarat, SV0394 (CMUB, BR); ibid. 19°06’36.2” N–98°44’41.1” E, elev. 1080 m, 23 May 2017, Santhiti Vadthanarat, SV0396 (CMUB, BR).

Notes—In the new species, the hymenophoral cystidia contained greenish yellow (1A8) pigments when fresh specimens were observed in water under a compound microscope. However, the pigment was discolored when the cystidia were observed in KOH or NH_4_OH, or after treatment of the specimen with heat (drying at 45–50 °C).

A macro-morphologically similar species, *Butyriboletus roseoflavus* (Hai B. Li & Hai L. Wei) D. Arora & J.L. Frank originally described from China, has a similar color tone of basidiomata with a light pink, light purplish red to rose-red pileus; and lemon-yellow, olive-yellow or honey-yellow hymenophore. However, it can be differentiated from *R*. *auriporus* by having a yellower and reticulated stipe which is lemon-yellow or light yellow with almost entirely reticulate stipe or at least in lower part; yellower context which is lemon-yellow and also variable staining reaction in response to bruising, bruising blue slowly or unchanging; bruising blue promptly hymenophore; subfusiform basidiospores; olive brown spore deposit; and the habitat in *Pinus* or mixed forests dominated by *Pinus* [52,53].

The chemical reaction of basidiospores with Melzer’s reagent which was negative to weakly dextrinoid in *R*. *auriporus* is also present in two Xerocomoideae species, *Alessioporus ichnusanus* (Alessio, Galli & Littini) Gelardi, Vizzini & Simonini, and *Pulchroboletus roseoalbidus* (Alessio & Littini) Gelardi, Vizzini & Simonini. However, the two species are different from *R*. *auriporus* by their basidiospore shapes, which are sub-cylindrical or ellipsoid or ellipsoid-fusoid, the strong discoloration (bluing or darkening) in parts of basidiomata, and olive-brown spore deposit [54].

## 4. Discussion

The new genus *Rubinosporus* is distinguished from other Boletaceae by a combination of striking characters, i.e., a golden yellow tubular hymenophore that is relatively thin especially when young, and dark ruby spore deposits. The character of golden yellow tubular hymenophore is also found in *Aureoboletus*, *Alessioporus*, and *Pulchroboletus*, which also belong to the subfamily Xerocomoideae. However, *Aureoboletus* species differ from *Rubinosporus* in usually having a viscid pileus surface especially when moist, olive brown spore deposit, and subfusiform or oblong ovoid to subglobose basidiospores [3,16,31]. *Alessioporus* is clearly different by the reticulated stipe occasionally with a granular ring-like zone in the middle or lower half of the stipe; rapidly bluing hymenophore, stipe surface, and context when bruised or exposed; sub-ellipsoid to fusiform, ellipsoid to subcylindrical basidiospores; olive brown spore deposit; and distribution in Mediterranean Italy and subtropical USA [54,55]. *Pulchroboletus* differs by the stipe surface with scattered red to reddish brown punctae, occasionally with reticulum or longitudinal striations, and with a pseudo-annulus; hymenophore and context usually intensively staining blue when bruised or cut; ellipsoidal to ellipsoid-fusoid basidiospores; olive brown colored spore deposit; so far found in Mediterranean Europe and tropical to subtropical America [46,54,56].

The thin hymenophore is also present in *Baorangia* and *Lanmaoa* G. Wu & Zhu L. Yang, which both belong to the *Pulveroboletus* group. They have a very thin hymenophore, with a tube length 1/3 to 1/5 times the thickness of the pileus context. However, *Baorangia* and *Lanmaoa* differ from *Rubinosporus* by having yellow hymenium (not golden or bright yellow) that immediately turns light blue to greenish blue when touched; olive-brown spore deposit; and subfusiform to elongated subfusiform basidiospores [5,17,35].

Although the color of spore deposit in *Rubinosporus* is somewhat similar to the color tone in *Austroboletus* (Corner) Wolfe and *Ionosporus* Khmeln., *Austroboletus* has a rufous madder to chocolate to purplish vinaceous spore deposit which is browner than in *Rubinosporus*. Also, *Austroboletus* species produce basidiomata with pileipellis markedly exceeding pileus margin, embracing the stipe in young basidiomata, whitish to pinkish hymenium, and ornamented basidiospores [3,57,58]. *Ionosporus* has pale violet to reddish brown spore deposit. However, their basidiospores have an obvious reaction in potassium hydroxide solution, turning deep purple violet. The basidiospores also have granulose pitted surface under SEM. Moreover, *Austroboletus* and *Ionosporus* phylogenetically belong to different subfamilies, the Austroboletoideae and Leccinoideae, respectively [2,10]. The color of spore deposit is one of the important character used to differentiate mushroom genera, both in the Agaricales and Boletales. Several previous studies used this character to differentiate new genera. For example, *Tylopilus eximius* (Peck) Singer, which has a reddish-brown spore deposit, was separated from *Tylopilus* (having a pinkish spore deposit), and placed in a new genus, *Sutorius* [48]. *Cacaoporus* is distinguished from the most similar genus *Sutorius* by its dark brown spore deposit while the genus *Sutorius* has a reddish-brown spore deposit [11]. Moreover, they were all supported by the phylogenies.

Most morphological characters of *Rubinosporus* fit the typical characters of Xerocomoideae genera, as described in Wu et. al. [2,3] and Zhu et al. [15]. However, the color of the spore deposit, which in all Xerocomoideae so far described has an olive-brown tint, is dark ruby in *Rubinosporus*. The differences in spore deposit color between genera within the same subfamily in Boletaceae, are also found in Boletoideae which varies from olive green (*Boletus*), blackish brown (*Strobilomyces*), light yellow or yellow golden (*Xanthoconium*), pinkish (*Tylopilus*) e.g., [2,3,59,60]. Spore print color, however, is mostly conserved at genus level, with only slight variations. Most Xerocomoideae genera, i.e., *Boletellus*, *Hemileccinum*, *Heimioporus*, *Hourangia*, *Phylloporus*, *Xerocomus* and some species in *Aureoboletus* produce ornamented basidiospores [3,15,40,61,62,63,64]. Exceptions exist, however, e.g., in *Phylloporus* and *Xerocomus* [44]. Smooth basidiospores are found in *Alessioporus*, *Pulchroboletus*, most species in *Aureoboletus* [3,31,46,54,55], and the new genus *Rubinosporus*. The basidiospores of the single *Rubinosporus* species, *R. auriporus* showed a weakly dextrinoid reaction in Melzer’s reagent, similar to two species in *Alessioporus* and *Pulchroboletus*, namely *A*. *ichnusanus* and *P*. *roseoalbidus* whereas the other species in the two latter genera are not dextrinoid [46,54,55,56]. Therefore, the character cannot be considered typical for *Rubinosporus*.

In the Xerocomoideae-wide phylogeny obtained in this study, the monophyly of all genera was highly supported. However, no sequences of *Alessioporus* were added in the phylogeny because among the genes that were used to infer our phylogeny, only a partial tef1 sequence of *A*. *ichnusanus* was available in GenBank. However, in ITS and combined ITS+ LSU+*tef*1 phylogenies of previous studies, *Alessioporus* was sister to *Pulchroboletus* [46,54,56]. In this study, *Pulchroboletus* was sister to *Aureoboletus* with high support, and distant from *Rubinosporus*. Moreover, *Alessioporus* is morphologically clearly different from *Rubinosporus* as discussed above. The relationship of *Rubinosporus* to the other genera within Xerocomoideae remains unclear. It formed a clade close to *Hemileccinum* with poor support. More species, genes, phylogenies are needed to reveal the sister relationship of *Rubinosporus*. In addition, the three genera *Phylloporus*, *Hourangia*, and *Xerocomus* formed a highly supported clade. *Phylloporus* formed a clade sister to *Xerocomus*, and both genera are sisters to *Hourangia*. The result is slightly different from Wu et al. [3] phylogeny (based on 28S, *tef*1, *rpb*1, and *rpb*2), in which the three genera also formed a highly supported clade but *Phylloporus* was sister to *Hourangia*, not *Xerocomus* like in this study.

Most of the Boletaceae genera have been recognized as important ectomycorrhizal fungi in forest ecosystems [65,66]. *Rubinosporus* also presumably forms ectomycorrhizal relationships with either Dipterocarpaceae or Fagaceae, or both. These two tree families were dominant around the area where the genus was found. However, further research is needed to confirm the ectomycorrhizal host species of *Rubinosporus*.

*Rubinosporus* is the third novel bolete genus described from Thailand, after *Spongiforma* Desjardin, Manfr. Binder, Roekring & Flegel, and *Cacaoporus* Raspé & Vadthanarat were described in 2009 and 2019, respectively [11,67]. Prior to this study, Boletaceae subfamily Xerocomoideae consisted of nine genera [2,3]. Based on the morphological and phylogenetic results in the present study, the tenth genus, *Rubinosporus* is introduced in the subfamily Xerocomoideae.

## Figures and Tables

**Figure 1 jof-08-00278-f001:**
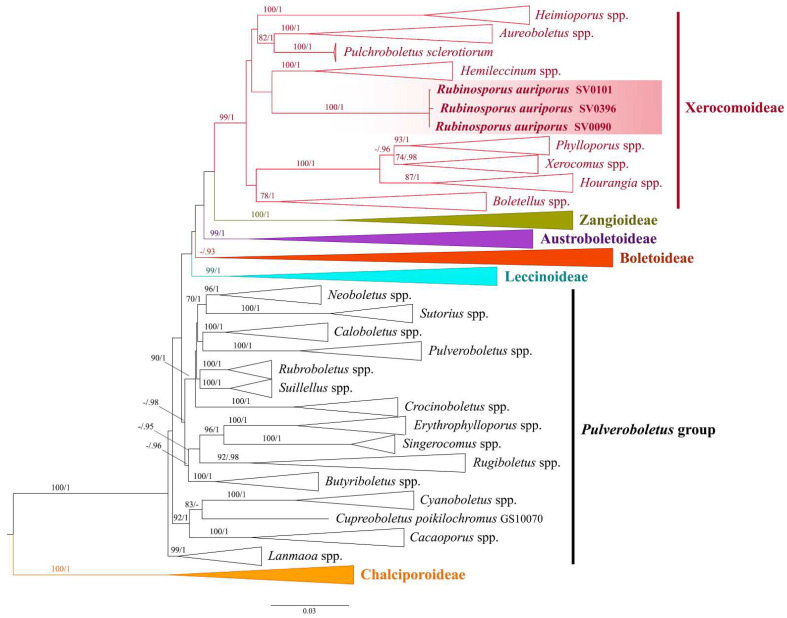
Boletaceae-wide Maximum Likelihood phylogenetic tree inferred from the four-gene dataset (*atp*6, *cox*3, *rpb*2, and *tef*1) (introns excluded), showing position of the new genus *Rubinosporus* in Xerocomoideae. Bootstrap support values (BS ≥ 70%) and the corresponding Bayesian posterior probabilities (PP ≥ 0.90) are shown above the supported branches. The two *Buchwaldoboletus* and seven *Chalciporus* species (subfamily Chalciporoideae) were used as the outgroup. All taxa belonging to subfamilies Austroboletoideae, Boletoideae, Chalciporoideae, Leccinoideae, and Zangioideae were collapsed into subfamily clades. All generic clades in subfamily Xerocomoideae and *Pulveroboletus* group that were highly supported were also collapsed.

**Figure 2 jof-08-00278-f002:**
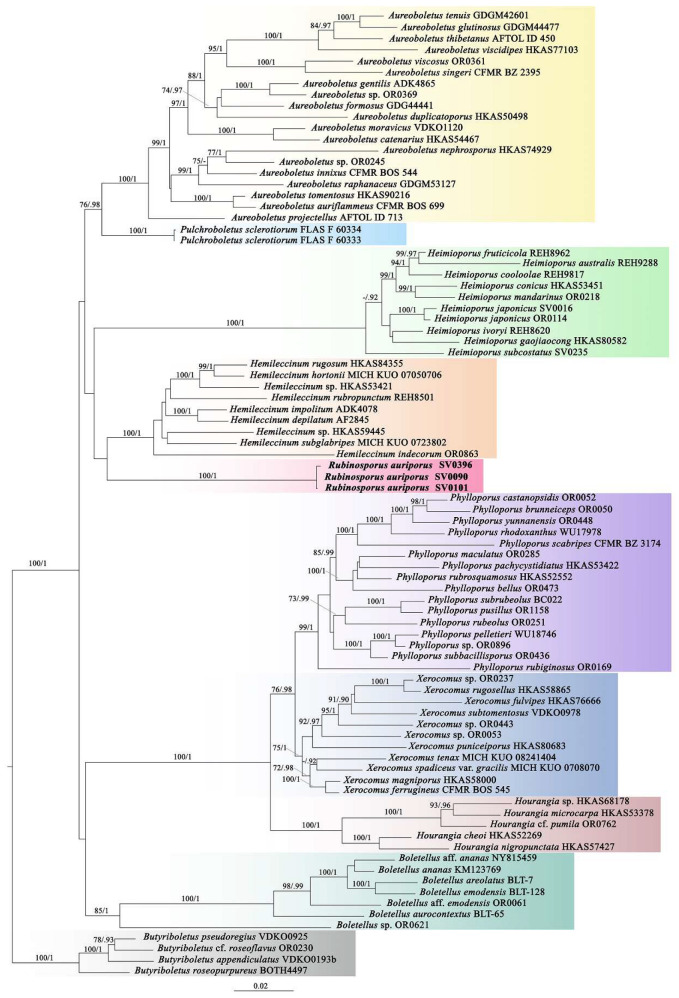
Xerocomoideae-wide phylogenetic tree inferred from the four-gene dataset (*atp*6, *cox*3, *rpb*2, and *tef*1) (introns included), including new genus *Rubinosporus* and selected Xerocomoideae using Maximum Likelihood and Bayesian Inference methods (ML tree is presented). The four *Butyriboletus* species in *Pulveroboletus* group were used as the outgroup. Bootstrap support values (BS ≥ 70%) and posterior probabilities (PP ≥ 0.90) are shown above the supported branches.

**Figure 3 jof-08-00278-f003:**
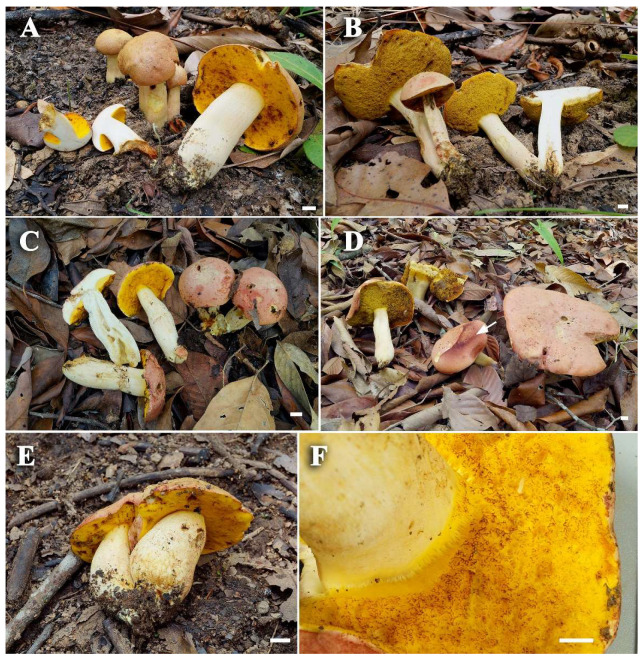
Fresh basidiomata of Rubinosporus auriporus: (**A**,**B**) SV0090 (Holotype); (**C**,**D**) SV0394, spores deposit on the cap showing dark ruby color (white arrow); (**E**) SV0396; (**F**) the golden yellow pores, irregularly reddish brown at places in (SV0394)—Bars (**A**–**E**) = 1, (**F**) = 5 mm.

**Figure 4 jof-08-00278-f004:**
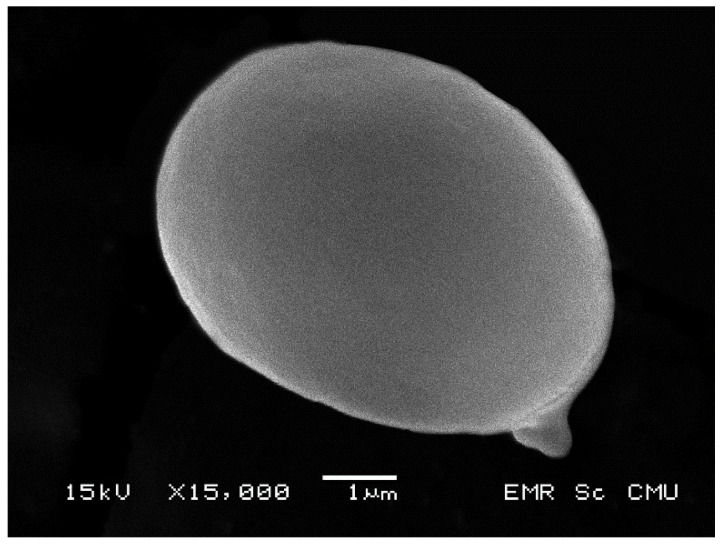
Scanning electron micrographs of Rubinosporus auriporus basidiospores from the holotype—Bar = 1 µm.

**Figure 5 jof-08-00278-f005:**
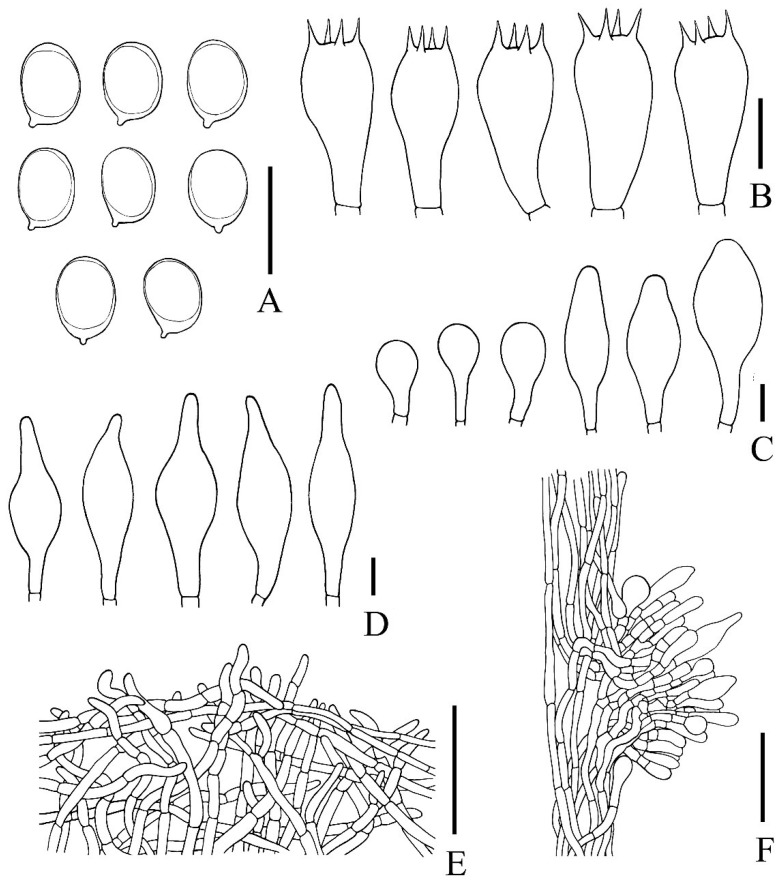
Microscopic features of Rubinosporus auriporus: (**A**) Basidiospores; (**B**) Basidia; (**C**) Two shapes of cheilocystidia; (**D**) Caulocystidia; (**E**) Pileipellis; (**F**) Stipitipellis.—Bars A–D = 10 µm, E,F = 50 µm. All drawings were made from the type (SV0090).

**Table 1 jof-08-00278-t001:** List of collections used for phylogenetic analyses, with origin, GenBank accession numbers, and reference(s).

Species	Voucher	Origin	*atp*6	*cox*3	*rpb*2	*tef*1	Reference(s)
*Afroboletus* aff. *multijugus*	JD671	Burundi	MH614651	MH614794	MH614747	MH614700	[11]
*Afroboletus costatisporus*	ADK4644	Togo	KT823958	MH614795 *	KT823991	KT824024	[21], [11] *
*Afroboletus luteolus*	ADK4844	Togo	MH614652	MH614796	MH614748	MH614701	[11]
*Aureoboletus auriflammeus*	CFMR:BOS-699	USA	–	–	MK766269	MK721060	[29]
*Aureoboletus catenarius*	HKAS54467	China	–		KT990349	KT990711	[3]
*Aureoboletus duplicatoporus*	HKAS50498	China	–	–	KF112754	KF112230	[2]
*Aureoboletus formosus*	GDGM44441	China	–	–	KT291751	KT291744	[30]
*Aureoboletus gentilis*	ADK4865	Belgium	KT823961	MH614797 *	KT823994	KT824027	[21], [11] *
*Aureoboletus glutinosus*	GDGM44477	China	–	–	MH700229	MH700205	[31]
*Aureoboletus innixus*	CFMR:BOS-544	USA	–	–	MK766270	MK721061	[29]
*Aureoboletus moravicus*	VDKO1120	Belgium	MG212528	MH614798 *	MG212615	MG212573	[32], [11] *
*Aureoboletus nephrosporus*	HKAS74929	China	–	–	KT990358	KT990721	[3]
*Aureoboletus projectellus*	AFTOL-ID-713	USA	DQ534604 *	–	AY787218	AY879116	[33] *, Unpublished
*Aureoboletus raphanaceus*	GDGM 53127	China	–	–	MN549706	MN549676	[31]
*Aureoboletus singeri*	CFMR:BOS-468	Belize	–	–	MK766274	MK721065	[29]
*Aureoboletus* sp.	OR0245	China	MH614653	MH614799	MH614749	MH614702	[11]
*Aureoboletus* sp.	OR0369	Thailand	MH614654	MH614800	MH614750	MH614703	[11]
*Aureoboletus tenuis*	GDGM42601	China	–	–	KT291754	KT291745	[30]
*Aureoboletus thibetanus*	AFTOL-ID-450	China	DQ534600 *	–	DQ366279	DQ029199	[33] *, Unpublished
*Aureoboletus tomentosus*	HKAS90216	China	–	–	KT990355	KT990717	[3]
*Aureoboletus viscidipes*	HKAS77103	China	–	–	KT990360	KT990723	[3]
*Aureoboletus viscosus*	OR0361	Thailand	MH614655	MH614801	MH614751	MH614704	[11]
*Australopilus palumanus*	REH-9433	Australia	–	–	MK766276	MK721067	[29]
*Austroboletus* cf. *dictyotus*	OR0045	Thailand	KT823966	MH614802 *	KT823999	KT824032	[21], [11] *
*Austroboletus* cf. *subvirens*	OR0573	Thailand	MH614656	MH614803	MH614752	MH614705	[11]
*Austroboletus olivaceoglutinosus*	HKAS57756	China	–	–	KF112764	KF112212	[2]
*Austroboletus* sp.	OR0891	Thailand	MH614657	MH614804	MH614753	MH614706	[11]
*Boletellus* aff. *ananas*	NY815459	Costa Rica	–	–	KF112760	KF112308	[2]
*Boletellus* aff. *emodensis*	OR0061	Thailand	KT823970	MH614806 *	KT824003	KT824036	[21], [11] *
*Boletellus ananas*	K(M)123769	Belize	MH614658	MH614807	MH614754	MH614707	[11]
*Boletellus areolatus*	TNS-F-61444	Japan	–	AB989025	AB999754	–	[34]
*Boletellus aurocontextus*	TNS-F-61501	Japan	–	AB989037	AB999770	–	[34]
*Boletellus emodensis*	TNS-F-61564	Japan	–	AB989053	AB999782	–	[34]
*Boletellus* sp.	OR0621	Thailand	MG212529	MH614808 *	MG212616	MG212574	[32], [11] *
*Boletus aereus*	VDKO1055	Belgium	MG212530	MH614809 *	MG212617	MG212575	[32], [11] *
*Boletus albobrunnescens*	OR0131	Thailand	KT823973	MH614810 *	KT824006	KT824039	[21], [11] *
*Boletus edulis*	VDKO0869	Belgium	MG212531	MH614811 *	MG212618	MG212576	[32], [11] *
*Boletus rubriceps*	MICH:KUO-08150719	USA	–	–	MK766284	MK721076	[29]
*Boletus* s.s. sp.	OR0446	China	MG212532	MH614813 *	KF112703	MG212577	[32], [11] *
*Borofutus dhakanus*	OR0345	Thailand	MH614660	MH614814	MH614755	MH614709	[11]
*Buchwaldoboletus lignicola*	HKAS76674	China	–	–	KF112819	KF112277	[2]
*Buchwaldoboletus lignicola*	VDKO1140	Belgium	MH614661	MH614815	MH614756	MH614710	[11]
*Butyriboletus appendiculatus*	VDKO0193b	Belgium	MG212537	MH614816 *	MG212624	MG212582	[32], [11] *
*Butyriboletus* cf. *roseoflavus*	OR0230	China	KT823974	MH614819 *	KT824007	KT824040	[21], [11] *
*Butyriboletus floridanus*	BOS-617	Belize	–	–	MK766287	MK721079	[29]
*Butyriboletus frostii*	NY815462	USA	–	–	KF112675	KF112164	[2]
*Butyriboletus pseudoregius*	VDKO0925	Belgium	MG212538	MH614817 *	MG212625	MG212583	[32], [11] *
*Butyriboletus roseopurpureus*	BOTH4497	USA	MG897418	MH614818 *	MG897438	MG897428	[35], [11] *
*Butyriboletus subsplendidus*	HKAS50444	China	–	–	KT990379	KT990742	[3]
*Butyriboletus yicibus*	HKAS55413	China	–	–	KF112674	KF112157	[2]
*Cacaoporus pallidicarneus*	SV0221	Thailand	MK372262	MK372299	MK372286	MK372273	[11]
*Cacaoporus tenebrosus*	SV0223	Thailand	MK372266	MK372303	MK372290	MK372277	[11]
*Caloboletus calopus*	ADK4087	Belgium	MG212539	MH614820	KP055030	KJ184566	[32], [36], [37], [11]
*Caloboletus firmus*	BOS-372	Belize	–	–	MK766288	MK721080	[29]
*Caloboletus inedulis*	BOTH3963	USA	MG897414	MH614821 *	MG897434	MG897424	[35], [11] *
*Caloboletus radicans*	VDKO1187	Belgium	MG212540	MH614822 *	MG212626	MG212584	[32], [11] *
*Caloboletus* sp.	OR0068	Thailand	MH614662	MH614823	MH614757	MH614711	[11]
*Caloboletus yunnanensis*	HKAS69214	China	–	–	KT990396	KJ184568	[36], [3]
*Chalciporus* aff. *piperatus*	OR0586	Thailand	KT823976	MH614824 *	KT824009	KT824042	[21], [11] *
*Chalciporus* aff. *rubinus*	OR0139	China	MH614663	–	MH614758	MH614712	[11]
*Chalciporus africanus*	JD517	Cameroon	KT823963	MH614825 *	KT823996	KT824029	[21], [11] *
*Chalciporus piperatus*	VDKO1063	Belgium	MH614664	MH614826	MH614759	MH614713	[11]
*Chalciporus rubinus*	AF2835	Belgium	KT823962	–	KT823995	KT824028	[21]
*Chalciporus* sp.	OR0363	Thailand	MH645586	MH645607	MH645602	MH645594	[11]
*Chalciporus* sp.	OR0373	Thailand	MH645587	MH645608	MH645603	MH645595	[11]
*Chamonixia brevicolumna*	DBG_F28707	USA	–	–	MK766291	MK721083	[29]
*Chamonixia caespitosa*	OSC117571	USA	–	–	MK766293	MK721085	[29]
*Chiua* sp.	OR0141	China	MH614665	MH614827	MH614760	MH614714	[11]
*Chiua virens*	OR0266	China	MG212541	MH614828 *	MG212627	MG212585	[32], [11] *
*Chiua viridula*	HKAS74928	China	–	–	KF112794	KF112273	[2]
*Crocinoboletus* cf. *laetissimus*	OR0576	Thailand	KT823975	MH614833 *	KT824008	KT824041	[21], [11] *
*Crocinoboletus rufoaureus*	HKAS53424	China	–	–	KF112710	KF112206	[2]
*Cupreoboletus poikilochromus*	GS10070	Italy	–	–	KT157068	KT157072	[38]
*Cyanoboletus brunneoruber*	OR0233	China	MG212542	MH614834 *	MG212628	MG212586	[32], [11] *
*Cyanoboletus pulverulentus*	RW109	Belgium	KT823980	MH614835 *	KT824013	KT824046	[21], [11] *
*Cyanoboletus sinopulverulentus*	HKAS59609	China	–	–	KF112700	KF112193	[2]
*Cyanoboletus* sp.	OR0257	China	MG212543	MH614836 *	MG212629	MG212587	[32], [11] *
*Cyanoboletus* sp.	OR0322	Thailand	MH614673	MH614837	MH614768	MH614722	[11]
*Cyanoboletus* sp.	OR0961	Thailand	MH614675	MH614839	MH614770	MH614724	[11]
*Erythrophylloporus aurantiacus*	REH7271	Costa Rica	MH614666	MH614829	MH614761	MH614715	[39]
*Erythrophylloporus fagicola*	Garay215	Mexico	MH614667	MH614830	MH614762	MH614716	[39]
*Erythrophylloporus paucicarpus*	OR1151	Thailand	MH614670	MH614831	MH614765	MH614719	[39]
*Erythrophylloporus suthepense*	SV0236	Thailand	MH614672	MH614832	MH614767	MH614721	[39]
*Fistulinella prunicolor*	REH9880	Australia	MH614676	MH614840	MH614771	MH614725	[11]
*Harrya chromapes*	HKAS50527	China	–	–	KF112792	KF112270	[2]
*Harrya moniliformis*	HKAS49627	China	–	–	KT990500	KT990881	[3]
*Heimioporus conicus*	HKAS53451	China	–	–	KF112805	KF112226	[3]
*Heimioporus australis*	REH9288	Australia	–	–	–	KP327703	[40]
*Heimioporus cooloolae*	REH9817	Australia	–	–	–	KP327710	[40]
*Heimioporus fruticicola*	REH8962	Australia	–	–	–	KP327696	[40]
*Heimioporus gaojiaocong*	HKAS80582	China	–	–	KT990409	KT990770	[3]
*Heimioporus ivoryi*	REH8620	Costa Rica	–	–	–	KP327683	[40]
*Heimioporus japonicus*	OR0114	Thailand	KT823971	–	KT824004	KT824037	[21]
*Heimioporus japonicus*	SV0016	Thailand	MT136776	–	MT136766	MT136771	[41]
*Heimioporus mandarinus*	OR0218	Thailand	MG212546	–	MG212632	MG212590	[32]
*Heimioporus subcostatus*	SV0235	Thailand	MT136780	–	MT136770	MT136775	[41]
*Hemileccinum depilatum*	AF2845	Belgium	MG212547	MH614843 *	MG212633	MG212591	[32], [11] *
*Hemileccinum hortonii*	MICH:KUO-07050706	USA	–	–	MK766377	MK721175	[29]
*Hemileccinum impolitum*	ADK4078	Belgium	MG212548	MH614844 *	MG212634	MG212592	[32], [11] *
*Hemileccinum indecorum*	OR0863	Thailand	MH614677	MH614845	MH614772	MH614726	[11]
*Hemileccinum rubropunctum*	REH-8501	USA	–	–	MK766327	MK721122	[29]
*Hemileccinum rugosum*	HKAS84355	China	–	–	KT990413	KT990774	[3]
*Hemileccinum* sp.	HKAS59445	China	–	–	KT990414	KT990775	[3]
*Hemileccinum* sp.	HKAS53421	China	–	–	KF112751	KF112235	[2]
*Hemileccinum subglabripes*	MICH:KUO-07230802	USA	–	–	MK766300	MK721092	[29]
*Hortiboletus amygdalinus*	HKAS54166	China	–	–	KT990416	KT990777	[3]
*Hortiboletus campestris*	MICH:KUO-08240502	USA	–	–	MK766302	MK721094	[29]
*Hortiboletus rubellus*	VDKO0403	Belgium	MH614679	MH614847	MH614774	–	[11]
*Hortiboletus subpaludosus*	HKAS59608	China	–	–	KF112696	KF112185	[2]
*Hourangia* cf. *pumila*	OR0762	Thailand	MH614680	MH614848	MH614775	MH614728	[11]
*Hourangia cheoi*	HKAS52269	China	–	–	KF112773	KF112286	[15]
*Hourangia microcarpa*	HKAS53378	China	–	–	KF112775	KF112300	[2]
*Hourangia nigropunctata*	HKAS 57427	China	–	–	KP136978	KP136927	[15]
*Hourangia* sp.	HKAS68178	China	–	–	KF112776	KF112301	[2]
*Hymenoboletus luteopurpureus*	HKAS46334	China	–	–	KF112795	KF112271	[2]
*Imleria badia*	VDKO0709	Belgium	KT823983	MH614849 *	KT824016	KT824049	[21], [11] *
*Imleria obscurebrunnea*	OR0263	China	MH614681	MH614850	MH614776	MH614729	[11]
*Imleria pallidus*	BOTH4356	USA	MH614659	MH614812	–	MH614708	[11]
*Lanmaoa angustispora*	HKAS74752	China	–	–	KM605177	KM605154	[17]
*Lanmaoa asiatica*	OR0228	China	MH614682	MH614851	MH614777	MH614730	[11]
*Lanmaoa carminipes*	BOTH4591	USA	MG897419	MH614852 *	MG897439	MG897429	[35], [11] *
*Lanmaoa pallidorosea*	BOTH4432	USA	MG897417	MH614853 *	MG897437	MG897427	[35], [11] *
*Lanmaoa* sp.	OR0130	Thailand	MH614683	MH614854	MH614778	MH614731	[11]
*Lanmaoa* sp.	OR0370	Thailand	MH614684	MH614855	MH614779	MH614732	[11]
*Leccinellum* aff. *crocipodium*	HKAS76658	China	–	–	KF112728	KF112252	[2]
*Leccinellum* aff. *griseum*	KPM-NC-0017832	Japan	KC552164	–	–	JN378450*	unpublished, [42] *
*Leccinellum cremeum*	HKAS90639	China	–	–	KT990420	KT990781	[3]
*Leccinum scabrum*	VDKO0938	Belgium	MG212549	MH614858 *	MG212635	MG212593	[32], [11] *
*Leccinum schistophilum*	VDKO1128	Belgium	KT823989	MH614859 *	KT824022	KT824055	[21], [11] *
*Leccinum variicolor*	VDKO0844	Belgium	MG212550	MH614860 *	MG212636	MG212594	[32], [11] *
*Mucilopilus castaneiceps*	HKAS75045	China	–	–	KF112735	KF112211	[2]
*Mycoamaranthus cambodgensis*	SV0197	Thailand	MZ355900	MZ355909	–	–	This study
*Neoboletus brunneissimus*	OR0249	China	MG212551	MH614861 *	MG212637	MG212595	[32], [11] *
*Neoboletus ferrugineus*	HKAS77718	China	–	–	KT990431	KT990789	[3]
*Neoboletus flavidus*	HKAS59443	China	–	–	KU974144	KU974136	[3]
*Neoboletus hainanensis*	HKAS59469	China	–	–	KF112669	KF112175	[2]
*Neoboletus junquilleus*	AF2922	France	MG212552	MH614862 *	MG212638	MG212596	[32], [11] *
*Neoboletus magnificus*	HKAS74939	China	–	–	KF112653	KF112148	[2]
*Neoboletus obscureumbrinus*	OR0553	Thailand	MK372271	–	MK372294	MK372282	[11]
*Neoboletus* sp.	OR0128	Thailand	MH614686	MH614863	MH614781	MH614734	[11]
*Neoboletus tomentulosus*	HKAS53369	China	–	–	KF112659	KF112154	[2]
*Neoboletus erythropus*	VDKO0690	Belgium	KT823982	MH614864 *	KT824015	KT824048	[21], [11] *
*Octaviania asterosperma*	AQUI3899	Italy	KC552159	–	–	KC552093	[43]
*Octaviania cyanescens*	PNW-FUNGI-5603	USA	KC552160	–	–	JN378438	[43], [42]
*Octaviania decimae*	KPM-NC17763	Japan	KC552145	–	–	JN378409	[43], [42]
*Octaviania tasmanica*	MEL2128484	Australia	KC552157	–	–	JN378437	[43], [42]
*Octaviania zelleri*	MES270	USA	KC552161	–	–	JN378440	[43], [42]
*Phylloporus bellus*	OR0473	China	MH580778	MH614866 *	MH580818	MH580798	[44], [11] *
*Phylloporus brunneiceps*	OR0050	Thailand	KT823968	MH614867 *	KT824001	KT824034	[21], [11] *
*Phylloporus castanopsidis*	OR0052	Thailand	KT823969	MH614868 *	KT824002	KT824035	[21], [11] *
*Phylloporus maculatus*	OR0285	China	MH580780	–	MH580820	MH580800	[44]
*Phylloporus pachycystidiatus*	HKAS53422	China	–	–	KF112777	KF112288	[2]
*Phylloporus pelletieri*	WU18746	Austria	MH580781	MH614869 *	MH580821	MH580801	[44], [11] *
*Phylloporus pusillus*	OR1158	Thailand	MH580783	MH614870 *	MH580823	MH580803	[44], [11] *
*Phylloporus rhodoxanthus*	WU17978	Austria	MH580785	MH614871 *	MH580824	MH580805	[44], [11] *
*Phylloporus rubeolus*	OR0251	China	MH580786	MH614872 *	MH580825	MH580806	[44], [11] *
*Phylloporus rubiginosus*	OR0169	China	MH580788	MH614873 *	MH580827	MH580808	[44], [11] *
*Phylloporus rubrosquamosus*	HKAS52552	China	–	–	KF112780	KF112289	[2]
*Phylloporus scabripes*	CFMR:BOS-621	Belize	–	–	MK766359	MK721156	[29]
*Phylloporus* sp.	OR0896	Thailand	MH580790	MH614874 *	MH580829	MH580810	[44], [11] *
*Phylloporus subbacillisporus*	OR0436	China	MH580792	MH614875 *	MH580831	MH580812	[44], [11] *
*Phylloporus subrubeolus*	BC022	Thailand	MH580793	MH614876 *	MH580832	MH580813	[44], [11] *
*Phylloporus yunnanensis*	OR0448	China	MG212554	MH614877 *	MG212640	MG212598	[32], [11] *
*Porphyrellus castaneus*	OR0241	China	MG212555	MH614878 *	MG212641	MG212599	[32], [11] *
*Porphyrellus* aff. *nigropurpureus*	ADK3733	Benin	MH614687	MH614879	MH614782	MH614735	[11]
*Porphyrellus nigropurpureus*	HKAS74938	China	–	–	KF112763	KF112246	[2]
*Porphyrellus porphyrosporus*	MB97 023	Germany	DQ534609	–	GU187800	GU187734	[33], [45]
*Porphyrellus* sp.	JD659	Burundi	MH614688	MH614880	MH614783	MH614736	[11]
*Porphyrellus* sp.	OR0222	Thailand	MH614689	MH614881	MH614784	MH614737	[11]
*Pulchroboletus sclerotiorum*	FLAS F 60333	USA	–	–	MF614169	MF614167	[46]
*Pulchroboletus sclerotiorum*	FLAS F 60334	USA	–	–	MF614164	MF614165	[46]
*Pulveroboletus* aff. *ravenelii*	ADK4360	Togo	KT823957	MH614882 *	KT823990	KT824023	[21], [11] *
*Pulveroboletus* aff. *ravenelii*	ADK4650	Togo	KT823959	MH614883 *	KT823992	KT824025	[21], [11] *
*Pulveroboletus brunneopunctatus*	HKAS55369	China	–	–	KT990455	KT990814	[3]
*Pulveroboletus fragrans*	OR0673	Thailand	KT823977	MH614884 *	KT824010	KT824043	[21], [11] *
*Pulveroboletus ravenelii*	REH2565	USA	KU665635	MH614885 *	KU665637	KU665636	[21], [11] *
*Retiboletus* aff. *nigerrimus*	OR0049	Thailand	KT823967	MH614886 *	KT824000	KT824033	[21], [11] *
*Retiboletus brunneolus*	HKAS52680	China	–	–	KF112690	KF112179	[2]
*Retiboletus fuscus*	OR0231	China	MG212556	MH614887 *	MG212642	MG212600	[32], [11] *
*Retiboletus griseus*	MB03 079	USA	KT823964	MH614888 *	KT823997	KT824030	[21], [11] *
*Retiboletus kauffmanii*	OR0278	China	MG212557	MH614889 *	MG212643	MG212601	[32], [11] *
*Retiboletus nigerrimus*	HKAS53418	China	–	–	KT990462	KT990824	[3]
*Rhodactina himalayensis*	CMU25117	Thailand	MG212558	–	–	MG212602, MG212603	[32]
*Rhodactina rostratispora*	SV0170	Thailand	MG212560	–	MG212645	MG212605	[32]
*Rossbeevera cryptocyanea*	KPM-NC17843	Japan	KT581441	–	–	KC552072	[43]
*Rossbeevera griseovelutina*	TNS-F-36989	Japan	KC552124	–	–	KC552076	[43]
*Rossbeevera pachydermis*	KPM-NC23336	New Zealand	KJ001064	–	–	KP222912	[43]
*Royoungia rubina*	HKAS53379	China	–	–	KF112796	KF112274	[2]
*Rubinosporus auriporus*	SV0090	Thailand	MZ355896	MZ355905	MZ355901	MZ355903	This study
*Rubinosporus auriporus*	SV0101	Thailand	MZ355897	MZ355906	MZ355902	MZ355904	This study
*Rubinosporus auriporus*	SV0394	Thailand	MZ355898	MZ355907	–	–	This study
*Rubinosporus auriporus*	SV0396	Thailand	MZ355899	MZ355908	–	–	This study
*Rubroboletus legaliae*	VDKO0936	Belgium	KT823985	MH614890 *	KT824018	KT824051	[21], [11] *
*Rubroboletus rhodosanguineus*	BOTH4263	USA	MG897416	MH614891 *	MG897436	MG897426	[35], [11] *
*Rubroboletus rhodoxanthus*	HKAS84879	China	–	–	KT990468	KT990831	[3]
*Rubroboletus satanas*	VDKO0968	Belgium	KT823986	MH614892 *	KT824019	KT824052	[21], [11] *
*Rugiboletus andinus*	REH-7705	Costa rica	–	–	MK766316	MK721111	[29]
*Rugiboletus brunneiporus*	HKAS83209	China	–	–	KM605168	KM605144	[17]
*Rugiboletus extremiorientalis*	OR0406	Thailand	MG212562	MH614893 *	MG212647	MG212607	[32], [11] *
*Singerocomus inundabilis*	TWH9199	Guyana	MH645588	MH645609	LC043089*	MH645596	[47] *, [11]
*Singerocomus rubriflavus*	TWH9585	Guyana	MH645589	MH645610	–	MH645597	[11]
*Spongiforma thailandica*	DED7873	Thailand	MG212563	MH614894 **	MG212648	KF030436*	[1] *, [32], [11] **
*Strobilomyces echinocephalus*	OR0243	China	MG212564	–	MG212649	MG212608	[32]
*Strobilomyces floccopus*	RW103	Belgium	KT823978	MH614895 *	KT824011	KT824044	[21], [11] *
*Strobilomyces mirandus*	OR0115	Thailand	KT823972	MH614896 *	KT824005	KT824038	[21], [11] *
*Strobilomyces* sp.	OR0259	China	MG212565	MH614897 *	MG212650	MG212609	[32], [11] *
*Strobilomyces* sp.	OR0319	Thailand	MH614690	MH614898	MH614785	MH614738	[11]
*Strobilomyces* sp.	OR0778	Thailand	MG212566	MH614899 *	MG212651	MG212610	[32], [11] *
*Strobilomyces* sp.	OR1092	Thailand	MH614691	MH614900	MH614786	MH614739	[11]
*Strobilomyces verruculosus*	HKAS55389	China	–	–	KF112813	KF112259	[2]
*Suillellus luridus*	VDKO0241b	Belgium	KT823981	MH614901 *	KT824014	KT824047	[21], [11] *
*Suillellus queletii*	VDKO1185	Belgium	MH645590	MH645611	MH645604	MH645598	[11]
*Suillellus subamygdalinus*	HKAS57262	China	–	–	KF112660	KF112174	[2]
*Sutorius australiensis*	REH9441	Australia	MG212567	MK386576 **	MG212652	JQ327032*	[48] *, [32], [11] **
*Sutorius eximius*	REH9400	USA	MG212568	MH614902 **	MG212653	JQ327029*	[48] *, [32], [11] **
*Sutorius pachypus*	OR0411	Thailand	MN067465	–	MN067500	MN067484	[49]
*Sutorius pseudotylopilus*	OR0378B	Thailand	MH614692	MH614903	MH614787	MH614740	[11]
*Sutorius rubinus*	OR0379	Thailand	MH614693	MH614904	MH614788	MH614741	[11]
*Sutorius ubonensis*	SV0032	Thailand	MN067472	–	MN067507	MN067491	[49]
*Tengioboletus glutinosus*	HKAS53425	China	–	–	KF112800	KF112204	[2]
*Tengioboletus reticulatus*	HKAS53426	China	–	–	KF112828	KF112313	[2]
*Turmalinea persicina*	KPM-NC18001	Japan	KC552130	–	–	KC552082	[43]
*Turmalinea yuwanensis*	KPM-NC18011	Japan	KC552138	–	–	KC552089	[43]
*Tylopilus balloui* s.l.	OR0039	Thailand	KT823965	MH614905 *	KT823998	KT824031	[21], [11] *
*Tylopilus felleus*	VDKO0992	Belgium	KT823987	MH614906 *	KT824020	KT824053	[21], [11] *
*Tylopilus ferrugineus*	BOTH3639	USA	MH614694	MH614907	MH614789	MH614742	[11]
*Tylopilus otsuensis*	HKAS53401	China	–	–	KF112797	KF112224	[2]
*Tylopilus* sp.	JD598	Gabon	MH614695	MH614908	MH614790	MH614743	[11]
*Tylopilus* sp.	OR0252	China	MG212569	MH614909 *	MG212654	MG212611	[32], [11] *
*Tylopilus* sp.	OR0542	Thailand	MG212570	MH614910 *	MG212655	MG212612	[32], [11] *
*Tylopilus* sp.	OR1009	Thailand	MH614697	MH614911	MH614791	–	[11]
*Tylopilus vinaceipallidus*	OR0137	China	MG212571	MH614912 *	MG212656	MG212613	[32], [11] *
*Tylopilus violaceobrunneus*	HKAS89443	China	–	–	KT990504	KT990886	[3]
*Veloporphyrellus conicus*	REH8510	Belize	MH614698	MH614913	MH614792	MH614745	[11]
*Veloporphyrellus gracilioides*	HKAS53590	China	–	–	KF112734	KF112210	[2]
*Veloporphyrellus pseudovelatus*	HKAS59444	China	JX984519	–	–	JX984553	[50]
*Veloporphyrellus velatus*	HKAS63668	China	JX984523	–	–	JX984554	[50]
*Xanthoconium affine*	NY00815399	USA	–	–	KT990486	KT990850	[3]
*Xanthoconium purpureum*	MICH:KUO-07061405	USA	–	–	MK766372	MK721170	[29]
*Xanthoconium sinense*	HKAS77651	China	–	–	KT990488	KT990853	[3]
*Xerocomellus chrysenteron*	VDKO0821	Belgium	KT823984	MH614914 *	KT824017	KT824050	[21], [11] *
*Xerocomellus cisalpinus*	ADK4864	Belgium	KT823960	MH614915 *	KT823993	KT824026	[21], [11] *
*Xerocomellus communis*	HKAS50467	China	–	–	KT990494	KT990858	[3]
*Xerocomellus ripariellus*	VDKO0404	Belgium	MH614699	MH614916	MH614793	MH614746	[11]
*Xerocomus ferrugineus*	CFMR:BOS-545	USA	–	–	MK766375	MK721173	[29]
*Xerocomus fulvipes*	HKAS76666	China	–	–	KF112789	KF112292	[2]
*Xerocomus magniporus*	HKAS58000	China	–	–	KF112781	KF112293	[2]
*Xerocomus puniceiporus*	HKAS80683	China	–	–	KU974146	KU974138	[3]
*Xerocomus rugosellus*	HKAS58865	China	–	–	KF112784	KF112294	[2]
*Xerocomus* s.s. sp.	OR0237	China	MH580796	–	MH580835	MH580816	[44]
*Xerocomus* s.s. sp.	OR0443	China	MH580797	MH614917 *	MH580836	MH580817	[44], [11] *
*Xerocomus* s.s. sp.	OR0053	Thailand	MH580795	MH614918 *	MH580834	MH580815	[44], [11] *
*Xerocomus spadiceus* var. *gracilis*	MICH:KUO-07080702	USA	–	–	MK766378	MK721176	[29]
*Xerocomus subtomentosus*	VDKO0987	Belgium	MG212572	MH614919 *	MG212657	MG212614	[32], [11] *
*Xerocomus tenax*	MICH:KUO-08241404	USA	–	–	MK766379	MK721177	[29]
*Zangia citrina*	HKAS52684	China	HQ326850	–	–	HQ326872	[51]
*Zangia olivaceobrunnea*	HKAS52272	China	HQ326857	–	–	HQ326876	[51]
*Zangia roseola*	HKAS51137	China	HQ326858	–	–	HQ326877	[51]

## Data Availability

Publicly available datasets were analyzed in this study. This data can be found here: (https://www.ncbi.nlm.nih.gov/; http://purl.org/phylo/treebase, submission ID 28349 and 28350; accessed on 1 December 2021).

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
