# Peer review of "Rubinosporus auriporus gen. et sp. nov. (Boletaceae: Xerocomoideae) from Tropical Forests of Thailand, Producing Unusual Dark Ruby Spore Deposits"

_jof, 2022, doi:10.3390/jof8030278_

Round 1

Reviewer 1 Report

Dear authors, congratulation for your work. The paper is clearly written, interesting, easy to read. 

line 26 correct "studied over a hundred years"

I would like to suggest to underline the role of this genus for nature, for science.

Author Response

Dear reviewer 1

Thank you very much for your valuable comments and suggestions on our manuscript. We sincerely appreciate all valuable comments and suggestions which helped us to improve the quality of the article. Appropriated changes, suggested by the Reviewer, have been applied to the manuscript

Our point-to-point response to the reviewer is as follows.

Reviewer 1 Comment 1: line 26 correct "studied over a hundred years"

Author repones: changed to “studied for over one hundred years”

Reviewer 1 Comment 2: I would like to suggest to underline the role of this genus for nature, for science.

Author repones: we have added a short paragraph to the discussion. The paragraph is as follows: “Most of Boletaceae genera have been recognized as important ectomycorrhizal fungi in forest ecosystems. Rubinosporus also presumably forms ectomycorrhizal relationships with either Dipterocarpaceae or Fagaceae, or both. These two tree families were dominant around the area where the genus was found. However, further research is needed to confirm the ectomycorrhizal host species of Rubinosporus.”

-----------------------------

Thank you very much

Best regards,

Author

Reviewer 2 Report

Here is the review titled "Rubinosporus auriporus gen. et sp. nov. (Boletaceae: Xerocomoideae) from tropical forests of Thailand, producing unusual dark ruby spore deposits" written by Santhiti Vadthanarat & co-authors.

The aim of the paper is to describe a new bolete genus Rubinosporus together with its type species R. auriporus based on collections found in tropical forests of Thailand. The genus has a dark ruby spore deposit which is unusual in Boletaceae. The phylogenetic analysis of the species in Boletaceae-wide dataset (based on four marker genes - atp6, cox3, rpb2, and tef1) showed that it belongs to an independent genus in subfamily Xerocomoideae. The new species/genus is described on the basis of its morphological and molecular characters which is accompanied with line drawings of microscopical elements of basidiomata.

The study is well done! Morphological description, phylogenetic study and discussion are exhaustive and cover all needed parts. The English language needs to be improved. The authors followed the newest version of International code of nomenclature for algae, fungi, and plants. There are a few additional minor points that need to be solved, so please check my reviewed version of the manuscript (pdf file).

The manuscript needs minor revision before publication in JoF.

Best, reviewer

Author Response

Dear reviewer 2,

Thank you very much for the review of our manuscript. We sincerely appreciate all valuable comments and suggestions which helped us to improve the quality of the article. Appropriated changes, suggested by the Reviewer, have been applied to the manuscript

Our point-to-point responses to the reviewer are in the attached file (.pdf).

Thank you very much

Best regards,

Author

Reviewer 3 Report

The development of molecular and computer techniques makes the recognition and description of new species increasingly easy.  This contributes significantly to the knowledge of a very diverse and not yet fully recognized world of fungi. The paper submitted for review concerns the newly described genus and species Rubinosporus auriporus from the tropical forests of Thailand. It is another article by the same team of authors who have been working on the diversity of boletales in Thailand for over 10 years. The manuscript is well thought out and very carefully prepared. The results of the study are well analyzed and the discussion is properly done. I consider the paper valuable and interesting.

Minor errors were found in the text:

In chapter 2.2. Morphological study and 3.1. Phylogenetic analyses delete spaces between numbers and “%”

Author Response

Dear reviewer 3,

Thank you very much for your valuable comments and suggestions on our manuscript. We sincerely appreciate all valuable comments and suggestions which helped us to improve the quality of the article. Appropriated changes, suggested by the Reviewer, have been applied to the manuscript

Our point-to-point response to the reviewer is as follows.

Reviewer 3 Comment 1: In chapter 2.2. Morphological study and 3.1. Phylogenetic analyses – delete spaces between numbers and “%”

Author repones: All spaces between number and “%” were removed.

-----------------------------------------

Thank you very much

Best regards,

Author